# Multi-Compartment, Early Disruption of cGMP and cAMP Signalling in Cardiac Myocytes from the *mdx* Model of Duchenne Muscular Dystrophy

**DOI:** 10.3390/ijms21197056

**Published:** 2020-09-25

**Authors:** Marcella Brescia, Ying-Chi Chao, Andreas Koschinski, Jakub Tomek, Manuela Zaccolo

**Affiliations:** Department of Physiology, Anatomy and Genetics, University of Oxford, Oxford OX1 3PT, UK; bresciam.dias@gmail.com (M.B.); inkychi@gmail.com (Y.-C.C.); andreas.koschinski@dpag.ox.ac.uk (A.K.); Jakub.tomek@dpag.ox.ac.uk (J.T.)

**Keywords:** Duchenne muscular dystrophy, cAMP, cGMP, NO, phosphodiesterases, cardiomyopathy, fluorescence resonance energy transfer, signalling compartmentalisation

## Abstract

Duchenne muscular dystrophy (DMD) is the most frequent and severe form of muscular dystrophy. The disease presents with progressive body-wide muscle deterioration and, with recent advances in respiratory care, cardiac involvement is an important cause of morbidity and mortality. DMD is caused by mutations in the dystrophin gene resulting in the absence of dystrophin and, consequently, disturbance of other proteins that form the dystrophin-associated protein complex (DAPC), including neuronal nitric oxide synthase (nNOS). The molecular mechanisms that link the absence of dystrophin with the alteration of cardiac function remain poorly understood but disruption of NO-cGMP signalling, mishandling of calcium and mitochondrial disturbances have been hypothesized to play a role. cGMP and cAMP are second messengers that are key in the regulation of cardiac myocyte function and disruption of cyclic nucleotide signalling leads to cardiomyopathy. cGMP and cAMP signals are compartmentalised and local regulation relies on the activity of phosphodiesterases (PDEs). Here, using genetically encoded FRET reporters targeted to distinct subcellular compartments of neonatal cardiac myocytes from the DMD mouse model *mdx*, we investigate whether lack of dystrophin disrupts local cyclic nucleotide signalling, thus potentially providing an early trigger for the development of cardiomyopathy. Our data show a significant alteration of both basal and stimulated cyclic nucleotide levels in all compartments investigated, as well as a complex reorganization of local PDE activities.

## 1. Introduction

Duchenne muscular dystrophy (DMD), one of the most common forms of muscular dystrophy, is caused by recessive mutations in the dystrophin gene on the X chromosome and affects 1 in 3500–5000 newborn males worldwide [1]. The disease impacts striated myocytes and is characterized by loss in skeletal muscle function leading to decreased ambulation. Until recently, respiratory failure secondary to neuromuscular dysfunction was the most frequent cause of death. However, improvements in respiratory care over the last few years have made cardiomyopathy (CM) an increasing cause of morbidity and mortality in DMD patients [2].

The dilated CM that typically associates with DMD results from a complex combination of pathological processes. While it is clear that the pathological alteration involves myocardial wasting and myocardial remodelling [3], the early molecular events that trigger the cardiac remodelling process remain incompletely understood.

Mutations associated with DMD result in absent or aberrant dystrophin [4]. In skeletal and cardiac muscle, dystrophin associates with several proteins forming the dystrophin-associated protein complex (DAPC). This complex localizes at the plasmalemma and links the actin cytoskeleton to the extracellular matrix, playing an important structural role. Lack of this complex disrupts sarcolemmal integrity and causes mechanical instability that results in easily damaged muscle fibers [5]. Components of the DAPC are also involved in non-structural functions, including modulation of intracellular calcium and redox homeostasis and the absence of dystrophin causes disrupted localisation and abnormal expression/activity of these proteins, leading to calcium mishandling and oxidative stress [6]. Increased vulnerability to mechanical stress is involved in the pathogenesis of dystrophic CM but secondary cellular responses involving dysregulation of ion fluxes at the plasmalemma, disruption of Ca^2+^ homeostasis, decreased nitric oxide (NO) production via neuronal nitric oxide synthase (nNOS) and disruption of mitochondrial signalling have also been implicated [7,8].

The second messengers, 3′-5′ cyclic guanosine monophosphate (cGMP) and 3′-5′ cyclic adenosine monophosphate (cAMP), regulate cardiac myocyte response to stress and both play a role in cardiac remodelling associated with dilated CM [9]. NO-activated soluble guanylyl cyclase (sGC) generates cGMP and mislocalisation and defective activity of nNOS has been documented both in humans with DMD [10,11] as well as in the *mdx* mouse [12], a well-established model of the disease. Reduced NO production and cGMP levels have been suggested to contribute to the CM associated to DMD, although the mechanisms remain unclear [13]. cAMP is generated by adenylyl cyclases (ACs) and mediates the adrenergic control of cardiac excitation-contraction coupling, modulates Ca^2+^ homeostasis, plasmalemma ion fluxes and mitochondrial function [14]. Early studies investigating the role of cAMP in skeletal muscle from DMD patients reported a defective response to catecholamines and suggested its involvement in the pathogenesis of the disease [15,16,17] but understanding of how cAMP signalling is affected in dystrophic cardiac myocytes is limited.

It is now well established that cardiac cyclic nucleotide signalling is compartmentalised and that differential spatial control of these second messengers is required for correct cardiac function [9,18]. The intracellular levels of cAMP and cGMP are regulated locally by phosphodiesterases (PDEs) [9]. This superfamily of cyclic nucleotide-degrading enzymes includes 11 families (PDE1–11) with selectivity for cAMP (PDE4, 7 and 8), cGMP (PDE5, 7 and 9) or with dual specificity (PDE1, 2, 3, 10 and 11) [19]. Different PDE isoforms have unique properties in terms of affinity and enzymatic kinetics, are differentially regulated and have distinct subcellular localization. By exerting locally their unique hydrolytic activity, PDEs fine-tune the level of cGMP and cAMP in a subcellular domain-specific manner, thus dictating activation of downstream targets. Disruption of the complex organization and regulation of subcellular cyclic nucleotide signalling domains can lead to cardiac disease [20]. No information is currently available on whether compartmentalized cyclic nucleotide signals are altered in DMD. Here we investigate the hypothesis that the structural changes that result from the lack of dystrophin may lead to disarray of local cyclic nucleotide signalling in dystrophic cardiac myocytes.

## 2. Results

To investigate whether the lack of dystrophin affects local cyclic nucleotide signals in the heart, we compared local cGMP and cAMP levels in cardiac myocytes from control mice (C57BL/10ScSnJ) and the *mdx* model (C57BL/10ScSn-Dmd < *mdx* > /J) which, as in human DMD, lacks dystrophin [21]. The *mdx* mouse develops a cardiac phenotype that becomes apparent at three months of age [22]. To avoid confounding effects on cyclic nucleotide signalling that may be secondary to the cardiac remodelling process rather than being a more direct consequence of the lack of dystrophin, we studied ventricular myocytes from neonatal mice (NVM). To monitor local cGMP and cAMP levels we used already available or newly developed fluorescence resonance energy transfer (FRET)-based reporters that are either distributed in the bulk cytosol or are targeted to the plasmalemma (PM) or to the outer mitochondrial membrane (OMM), three compartments that are thought to be relevant to the pathogenesis of DMD. For all targeted sensors used in this study, determination of Pearson’s correlation coefficient using appropriate markers shows selective targeting that is not affected in *mdx* cells compared to controls (Appendix A). A comparison of the dynamic range displayed by the cytosolic and targeted reporters used in this study as well as representative response kinetics are shown in Appendix A. In each compartment we assessed the basal level of cyclic nucleotide and the increase in response to stimulation. In addition, we used PDE family-selective inhibitors to manipulate local levels of cyclic nucleotides and to reveal differences in local regulation of cAMP and cGMP. In no case did the cyclic nucleotide increase resulting from application of the stimulus and of the PDE-selective inhibitor saturate the FRET reporters as in all experimental conditions and with all sensors, addition of the non-selective PDE inhibitor IBMX (for cGMP sensors) or a combination of IBMX and the AC activator forskolin (for the cAMP sensors) further increased the FRET signal, as can be appreciated in representative examples shown in Appendix A.

### 2.1. cGMP Signalling in the Cytosol

To measure cGMP in the bulk cytosol we used the previously described reporter cGi500 [23] (Figure 1A). As shown in Figure 1B, basal cGMP levels were significantly lower in *mdx* NVM compared to control, despite a larger amount of sGC was found to be expressed in NVM whole cell lysates from dystrophic compared to healthy myocytes (Appendix A). We next used cGMP-PDEs family-selective inhibitors to assess differences in the regulation of basal cGMP. Figure 1C shows that selective inhibition of PDE2A with Bay 60-7550 (Bay 60, 1 μM) resulted in a significantly larger increase in cGMP in control than in the *mdx* myocytes. In contrast, PDE5 inhibition with sildenafil (10 μM) yielded a small but detectable increase in cGMP in *mdx* cells but no effect was detectable in control cells. It should be noted that, given the high dose of sildenafil (10 μM) used in our experiments, the larger cGMP increase detected in *mdx* cells may be due, at least in part, to inhibition of PDE1 (IC_50_ of sildenafil for PDE1 = 100 nM [24]). No significant difference between control and *mdx* was recorded on inhibition of PDE3 (cilostamide, 10 μM) or PDE9 (PF044, 5 μM) (Figure 1C). We next stimulated synthesis of cGMP by treating NVM with Bay 41-2272 (Bay 41, 5 μM), a compound that stimulates sGC in a NO-independent manner, or with a combination of Bay 41 (5 μM) and the NO donor SNAP (50 μM) and found that these stimuli generate the same increase in cGMP in control and *mdx* myocytes (Figure 1D,E), suggesting that the ability of cytosolic sGC to synthesise cGMP is not compromised in *mdx* cells, provided sufficient NO is available. Selective inhibition of cGMP-PDEs in NVM pretreated with SNAP and Bay 41 did not show any significant difference between *mdx* and control (Figure 1F). The apparent discrepancy between the contribution of PDE2A in the regulation of basal and stimulated cGMP may be explained by the relatively low affinity (K_M_ = 10 μM) of PDE2A for cGMP [25]. While the response in the stimulated cells suggested equal amounts of PDE2A activity, the apparent lower contribution of PDE2A in basal conditions may derive from basal cGMP being lower in *mdx* cells than in controls and therefore insufficient to robustly engage PDE2A in hydrolysis. We also assessed PDE mRNA levels and found no significant differences between control and *mdx* NVM for any of the cGMP-PDEs tested (Appendix A).

### 2.2. cGMP Signalling at the Plasmalemma

Although the majority of sGC is found in the cytosol of cardiac myocytes, a significant fraction has been reported to localise at the plasmalemma [26] where the displacement of nNOS in *mdx* myocytes may affect local cGMP levels. To investigate cGMP levels at the plasmalemma we generated a version of cGi500 where a short peptide corresponding to the amino-terminal domain from the lyn kinase (PM) was fused to the amino terminus of the sensor (Figure 2A). This domain is post-translationally myristoylated and palmitoylated and targets the sensor to the plasma membrane [27] (Figure 2A). Similarly to what observed in the bulk cytosol, we found that basal levels of cGMP at the plasmalemma of *mdx* NVM are significantly lower than in controls (Figure 2B). At this site, the lower local cGMP in *mdx* cells may be partly due to the increased cGMP-PDE activity (Figure 2C). In contrast with what observed in the bulk cytosol, treatment with SNAP (50 µM) + Bay 41 (5 µM) attenuated but did not abolish the difference in cGMP between stimulated NVM from *mdx* and control (Figure 2D). Notably, when the myocytes were treated with 10 μM cinaciguat, a compound that, unlike NO, can also activate the oxidised form of sGC [28], we found that the amount of cGMP generated was significantly higher in *mdx* cells compared to control (Figure 2E), a response that was not recapitulated in the bulk cytosol (Appendix A). These data suggest that a larger fraction of the plasmalemma-associated sGC is oxidized in *mdx* compared to control myocytes and that this may contribute to the defective cGMP response that the dystrophic myocytes show at this site on activation. The data also indicate that the increased amount of sGC detected in *mdx* compared to control (Appendix A) may be largely represented by an oxidized form of the enzyme. Pharmacological inhibition of individual cGMP-PDEs in NVM pretreated with SNAP (50 µM) + Bay 41 (5 µM) resulted in similar increase in control and *mdx* myocytes (Figure 2F).

### 2.3. cGMP Signalling at the OMM

To measure cGMP levels at the OMM we fused a targeting peptide (T.70) from the yeast translocase protein TOM 70 [29] to the amino terminus of cGi500 to generate OMM-cGi500 (Figure 3A). FRET imaging showed no significant difference in basal cGMP between *mdx* and control NVM at the OMM (Figure 3B) and the contribution of individual cGMP-PDEs in regulating basal cGMP levels was similar in control and *mdx* cells, except for PDE2A, which did not appear to be engaged in significant hydrolysis of basal cGMP in dystrophic NVM (Figure 3C). The same FRET change was detected at the OMM in control and *mdx* NVM after activation of sCG with SNAP (50 µM) + Bay 41-2272 (5 μM), indicating similar increase in cGMP. For all cGMP-PDEs we found a similar contribution in *mdx* and control cells to the regulation of stimulated cGMP (Figure 3E).

### 2.4. cAMP Signalling in the Bulk Cytosol

We next investigated bulk cytosolic cAMP in control and *mdx* NVM using the cAMP FRET reporter EPAC-S^H187^ (H187) [30] (Figure 4A). We found significantly higher basal cAMP in in the cytosol of myocytes from *mdx* mice compared to control (Figure 4B). When family-selective cAMP-PDE inhibitors were applied to otherwise non-stimulated cells, we found that *mdx* NVM display significantly less PDE4 activity in the bulk cytosol compared to control (Figure 4C), which may explain the higher basal cAMP levels found in *mdx*. While PDE3 inhibition resulted in a small, but significantly larger increase in cAMP in *mdx* than in control, inhibition of PDE8 resulted in a larger cAMP increase in control compared to *mdx* (Figure 4C). No significant difference was detected on inhibition of PDE2A between the two cell types (Figure 4C). The different activity detected for some of the cAMP-PDEs could not be ascribed to a different level of gene transcription, as shown in Appendix A. NVM treated with the β-adrenergic receptor agonist isoproterenol (ISO, 1 nM) displayed a significantly lower cAMP response in *mdx* compared to control (Figure 4D). Inhibition of individual cAMP-PDE in NVM pre-treated with ISO showed that both PDE2A and PDE4 provide reduced cAMP hydrolytic activity in *mdx* compared to control when the cells are pre-treated with ISO (Figure 4E). The finding that in *mdx* cells ISO generates an attenuated cAMP response despite displaying a reduced cAMP-PDE activity suggests that *mdx* NVM may have reduced ability to synthesize cAMP upon activation of ACs. While analysis of mRNA levels for the two prevalent ACs, AC5 and AC6, showed no significant difference between *mdx* and control (Appendix A), we found that activation of all ACs with forskolin resulted in a significantly lower cAMP response in *mdx* compared to control (Figure 4F), compatible with disrupted enzymatic AC activity in the dystrophic myocytes [17,31].

### 2.5. cAMP Signalling at the Plasmalemma

To target the cAMP FRET reporter to the plasmalemma we applied the same strategy used for PM-cGi500 to generate PM-H187 (Figure 5A). FRET imaging of NVM expressing this sensor showed no significant difference in basal cAMP at the plasmalemma in *mdx* and control cells (Figure 5B). The basal activity of cAMP-PDEs was also similar in the two cell types, except for a significantly lower PDE8 contribution in *mdx* myocytes that may be balanced by a higher contribution of PDE4 (Figure 5C). On application of ISO 1 nM the same cAMP increase was detected in control and *mdx* cells (Figure 5D). No significant difference was found between the two cell types when cAMP-PDEs inhibitors were applied to cells pretreated with ISO (Figure 5E).

### 2.6. cAMP Signalling at the OMM

To investigate cAMP signals at the OMM we used the sensor OMM-H187 [32] (Figure 6A). At this site we found that basal cAMP is significantly lower in *mdx* compared to controls (Figure 6B). Such difference, in face of lower stimulated cAMP in the cytosol of *mdx* cells, may be explained by a significantly larger PDE8 hydrolytic activity at the OMM (Figure 6C) (see discussion). We also found a lower PDE2A-dependent hydrolysis of basal cAMP in *mdx* compared to control, which however may be due to the low affinity of PDE2 for cAMP (K_M_ = 30 μM) and the consequent minor engagement of PDE2A in hydrolysis of basal cAMP at this site. On ISO stimulation we found that at the OMM the cAMP response was apparently larger in *mdx* cells compared to control **(**Figure 6D), despite equal (for PDE2, PDE3 and PDE4) or larger (for PDE8) hydrolytic activity in dystrophic compared to healthy NVM (Figure 6E) (but see discussion).

## 3. Discussion

The aim of this study was to investigate whether the lack of dystrophin may affect compartmentalised cyclic nucleotide signalling in cardiac myocytes from the *mdx* model of DMD. We found profound alterations in both basal and stimulated levels of cGMP and cAMP in the bulk cytosol, the plasmalemma and at the OMM, as well as a complex reorganisation of PDE activities at these sites (Figure 7).

To study cyclic nucleotide levels in relevant subcellular compartments, we complemented existing FRET reporters with three new targeted sensors: PM-cGi500 and OMM-cGi500, to detect cGMP levels at the plasmalemma and mitochondria, respectively, and PM-H187, for detection of cAMP at the plasmalemma. We found that the localization of the newly developed targeted reporters is as expected and that it is not affected in *mdx* cells. To gauge the dynamic range of the newly generated reporters we measured the FET signal in the absence of cyclic nucleotide or in the presence of saturating concentration of cyclic nucleotide. As previously demonstrated for other FRET reporters [33], we found that fusion of a targeting sequence often affects the dynamic range of the sensor.

Our analysis show that lack of dystrophin in *mdx* NVM is associated with reduced cGMP levels, both in the bulk cytosol and at the plasmalemma. We find that stimulation of sCG with the NO-independent stimulator Bay 41, either alone or in combination with the NO donor SNAP, abolishes the difference in bulk cGMP between *mdx* and control cells, confirming that it is the reduced availability of NO that accounts for the deficit in cytosolic cGMP in the dystrophic cardiac myocytes. At the plasmalemma, a higher PDE activity and a more prominent fraction of oxidised sGC appear to contribute to defective cGMP levels. Lack of dystrophin is associated with deficit of nNOS activity [10,11] and the nNOS-cGMP signaling axis has emerged as a potential therapeutic target in DMD, although the mechanisms downstream of cGMP involved in the pathogenesis of the disease remain unclear. Defective NO-cGMP signalling in dystrophic myocytes results in unopposed sympathetic vasoconstriction, functional ischemia and cell injury. On the other hand, deranged NO-cGMP signalling within the myocyte can also be directly responsible for cell damage [34]. To complicate the picture, in dystrophic cardiac myocytes nNOS activity is reduced but eNOS is unchanged and iNOS activity is elevated, making the effects of NO-dependent physiology difficult to predict [3]. Our findings of a significant deficit of cGMP levels in dystrophic NVM, that is present early on and precede the development of a cardiac phenotype, suggest that alteration of cardiac myocyte cGMP signalling may be an early trigger of the CM associated to DMD.

In line with previous investigations [35] we find that the activity of cGMP-PDEs is elevated in *mdx* mice, although this appears to be restricted to the plasmalemma compartment. Previous studies in *mdx* mice showed that PDE5 inhibition with sildenafil improved cardiac performance [36]. Clinical trials to test the efficacy of PDE5 inhibitors in patients, however, failed to show beneficial effect [37,38]. In addition, although a tendency for sildenafil to reduce ROS was reported in experimental models, this was not associated with protective effects [39]. Here we find that PDE2A plays a more prominent role than other cGMP-PDEs in the regulation of cGMP, particularly in the bulk cytosol. Our data show that the stimulated cGMP response is significantly larger on inhibition of PDE2A than PDE5, both in control (*p* = 0.048) and in *mdx* myocytes (*p* = 0.0002). Our findings are consistent with the view that lack of therapeutic benefit with sildenafil may result from PDE5 inhibition producing only a modest increase in cGMP [3] and indicate PDE2A inhibition as an alternative therapeutic intervention, particularly in combination with NO donors/sGC activators [40].

NO-cGMP signalling has been implicated in mitochondrial biogenesis and respiration [41] and the improved cardiac performance observed in *mdx* mice treated with sildenafil has been proposed to involve a protective effect of the drug on mitochondria [42,43]. Interestingly, we found no significant difference in cGMP levels at the OMM in *mdx* compared to control, suggesting that a NO source in proximity of mitochondria [44] is not compromised in dystrophic cardiac myocytes and can supply regular levels of NO leading to normal concentrations of cGMP locally. Our findings may explain previous reports that sildenafil does not affect mitochondrial content or oxidative phosphorylation defects in *mdx* mice [45] and support a cGMP-independent mechanism for the mitochondrial alterations associated with DMD.

Early studies investigating cAMP signalling in human biopsies from DMD skeletal muscle reported defective basal and stimulated ACs activity and reduced PDE activity [17,31]. Here we show that in *mdx* NVM basal cAMP levels are differently affected in different subcellular compartments. In line with previous findings in skeletal muscle, we find that in dystrophic cells ACs display a reduced ability to synthesise cAMP, both in response to forskolin and upon activation of β-adrenergic receptors. Despite a reduced stimulated cyclase activity, we find that basal level of cAMP in the bulk cytosol is higher in *mdx* than in control, likely due to a significantly lower PDE4 hydrolytic activity in *mdx* in this compartment. On activation of β-adrenergic receptors, defective synthesis predominates over reduced hydrolysis, resulting in a cAMP response to ISO that is smaller in *mdx* than in control NVM. Our findings provide a possible mechanistic explanation for previous observations that young *mdx* mice display a hypercontractile phenotype but reduced ability to respond to adrenergic stimulation [46] as these may result from reduced cAMP-PDE (particularly PDE4) hydrolytic activity associated with a reduced ability of ACs to respond to activation.

We found no significant difference in basal or stimulated cAMP between control and *mdx* at the plasmalemma. In resting cells, this may be explained by a more prominent PDE4 activity at this site in *mdx* myocytes compared to controls. The near-identical response to ISO stimulation detected at the plasmalemma of the two cell types, however, appears to be in contradiction with the reduced stimulated ACs synthetic activity in *mdx* myocytes. One possible explanation is that in the plasmalemma compartment the cAMP diffusion rate is slower in *mdx* than in control myocytes. We have previously shown that the integrity of the cytoskeleton plays an important role in the diffusion of cAMP from the sub-membrane space to the bulk cytosol [47]. Dystrophin functions to mechanically anchor cytoplasmic γ-actin filaments of the cortical cytoskeleton with the sarcolemma in a well-defined and ordered pattern [48]. Lack of dystrophin leads to disarray of the sub-plasmalemma cytoskeleton [49] that may result in slower diffusion of cAMP that is trapped in a disordered cortical lattice.

We found significant differences between dystrophic myocytes and healthy controls at the OMM, both in basal and in stimulated cAMP. At the OMM, *mdx* NVM show a significantly lower resting cAMP level compared to control. The striking difference with the bulk cytosol, where *mdx* myocytes have significantly higher basal cAMP, can be explained by a different distribution of cAMP-PDEs at these two locations. In particular, our data with selective cAMP-PDE inhibitors indicate that *mdx* myocytes display significantly more PDE8 activity at the OMM and significantly less PDE4 activity in the bulk cytosol compared to controls. These PDEs have very distinct properties. PDE8 is a high affinity (K_M_ = 0.06 μM) and fast kinetics (V_max_ = 0.15 μmol × min^−1^ × mg^−1^) enzyme [50], whereas PDE4 is a relatively low affinity (K_M_ = 5 μM) and slow kinetics (V_max_ = 0.03 μmol × min^−1^ × mg^−1^) enzyme [51,52]. At low cAMP concentrations, as in basal conditions, PDE8 provides most of the cAMP hydrolytic activity at the OMM and this results in a lower basal cAMP in *mdx* cells. In the bulk cytosol, the reduced PDE4 activity in *mdx* myocytes allows higher basal cAMP levels compared to control cells. On β-adrenergic stimulation we find, counterintuitively, that the cAMP response at the OMM is higher in *mdx* than in control. Assuming that cAMP measured at the OMM diffuses from the cytosolic compartment, where the cAMP response to ISO is lower in *mdx* than in control, and considering that the OMM in *mdx* cells have more PDE8 activity, one would expect to see lower levels of stimulated cAMP at the OMM of *mdx* myocytes than in control cells. One possible explanation for this apparent incongruence is that, although the increase in cAMP is larger in *mdx* than in control, the actual cAMP concentration is still lower in *mdx* than in control myocytes. To test this hypothesis, we generated a mathematical model of the OMM compartment and populated it with PDE4 and PDE8, the two predominant cAMP-PDEs at this site (see Appendix A for further details) and simulated basal and stimulated cAMP levels. As shown in Appendix A, the model replicates our experimental finding that, on ISO stimulation, the cAMP change at the OMM is larger in *mdx* than in control. In addition, the model reveals that the cAMP concentration achieved at the OMM in *mdx* cells is lower than that achieved at this site in control cells (Appendix A), supporting our hypothesis. Moreover, the model recapitulates the experimental finding that basal cAMP levels at the OMM are lower in *mdx* myocytes compared to control, despite bulk cAMP being higher in *mdx* than in control. This is achieved only when a significantly larger amount of PDE8 activity is attributed to the OMM compartment compared to bulk cytosol of *mdx* cells, indicating that PDE8 plays a key role in determining the cAMP response at the OMM. Interestingly, our experimental results show that in *mdx* myocytes the larger amount of PDE8 activity at the OMM is accompanied by reduced PDE8 activity in the bulk cytosol and at the plasmalemma, suggesting a redistribution of PDE8 between compartments in dystrophic myocytes, an hypothesis that will need to be confirmed experimentally.

The altered cAMP level observed at the OMM in *mdx* myocytes is relevant as it may contribute to development and progression of the CM. It is well established that cAMP, via PKA, plays an important role in integrating redox signalling and in attenuating the detrimental impact of oxidative stress in cardiac myocytes [53] and the defective cAMP signalling we observe at the OMM in neonatal dystrophic myocytes may be an important contributor to cell damage. In addition, the cAMP/PKA axis drives mitochondrial respiration in exercise [54]. Our analysis indicates that both in unstimulated cells and upon ISO stimulation the cAMP response at the OMM is attenuated in *mdx*. This may lead to a limited ability of the mitochondria in dystrophic hearts to synthesise ATP, particularly in conditions of increased cardiac workloads [3]. Further investigations will be required to establish whether these cAMP-dependent mechanisms contribute to the CM associated with DMD and whether correction of cAMP signals at the mitochondria may be beneficial therapeutically.

## 4. Materials and Methods

### 4.1. Generation of Targeted FRET-Based Sensors

A DNA fragment encompassing amino acids MGCIKSKRKDNLNDD corresponding to the N-terminal targeting signal from Lyn Kinase (MP) [55] was inserted, by using specific restriction enzymes, upstream to CFP and mTurquoise2 in the cytosolic cGi500 and EPAC-S^H187^ (H187) FRET-based sensors for the generation of the plasma membrane targeted PM-cGi500 and PM-H187 reporters, respectively. A DNA fragment encompassing peptides of yeast translocase outer mitochondria membrane (OMM) protein TOM 70 (yTOM70) was extracted, via specific restriction enzyme digestion, from the OMM-H187 FRET-based sensor [32]. This fragment was inserted between specific restriction sites upstream of CFP in the cGi500 sensor backbone for the generation of the OMM-cGi500 FRET-based sensor. The adenoviral vectors were generated by Vector Biolabs (Vector Biolabs, Malvern, Pennsylvania, USA).

### 4.2. Determination of FRET Sensors Dynamic Range

Determination of FRET signals at zero and saturating concentration of cyclic nucleotide was performed either by saponin permeabilisation (cGi500 sensors), or microinfusion (H187 sensors). Microinfusion experiments were performed as described previously [56]. In brief, CHO cells expressing the specific FRET sensor were patch-clamped and simultaneously imaged under FRET imaging conditions (see below). After establishment of a tight seal between cell membrane and patch -pipette (“Gigaseal”) the whole-cell configuration was established. This configuration provides direct access from the pipette to the cytoplasm and cAMP can either diffuse from the pipette into the cell or vice versa, depending on the respective cAMP concentration in the patch-pipette solution. FRET-ratio changes were monitored for either 0 or 1 mM cyclic nucleotide in the patch pipette. Pipette resistance was in the range of 5–10 MΩ. Seal resistances typically were in the range of several Gigaohm, whole-cell resistances for CHO-cells were typically between 0.5 and 1 Gigaohm. Pipette solutions (“intracellular buffer”) contained 20 mM NaCl, 140 mM K-Glutamate, 2 mM MgCl_2_, 0.00404 mM CaCl_2_, 0.1 mM BAPTA (yielding a calculated free Ca^2+^ concentration of 10 nM, low buffered) and 10 mM HEPES. All solutions were adjusted with KOH according to the measured intracellular pH of the cells (here pH 7.55) and supplemented with cAMP as appropriate. The extracellular solution contained 140 mM NaCl, 3 mM KCl_3_, 2 mM MgCl_2_, 2 mM CaCl_2_, 15 mM Glucose, 10 mM HEPES and was adjusted with NaOH to pH 7.2. Electrophysiological data were acquired with a Cairn Optopatch patch-clamp amplifier (Cairn Research) controlled by WinWCP software (John Dempster, University of Strathclyde, Glasgow, Scotland, UK).

The saponin permeabilisation protocol was adapted from [57]. In brief, CHO cells were placed into the same intracellular buffer as described above except for BAPTA being replaced by 0.5 mM EGTA. The Ca^2+^-concentration was 0.16 µM, yielding a nominal free Ca^2+^-concentration of 10 nM, as calculated with MaxChelator (http://maxchelator.stanford.edu/). Imaging experiments were performed as described for the microinfusion experiments. After reaching a stable basal ratio, typically after about 3–5 min, saponin (from quillaja bark, Sigma, S4251, 8 µg/mL) alone or in combination with 1 mM cGMP, was applied and was permanently present. The resulting FRET change after saponin-induced membrane permeabilization was recorded. All FRET measurements for both microinfusion and saponin-permeabilisation experiments, were background-subtracted and, if necessary, corrected for baseline drifts. FRET changes were determined at steady state indicated by a plateau of the ratio change.

### 4.3. Neonatal Mouse Ventricular Cardiomyocyte Isolation

All procedures were carried out in compliance with the standards for the care and use of animal subjects as stated by the requirements of the UK Home Office (ASPA1986 Amendments Regulations 2012) incorporating the EU directive 2010/63/EU.

The isolation of neonatal mouse left ventricular cardiomyocytes was adapted from the neonatal rat left ventricular cardiomyocytes previously described [58]. In short, BL10 (C57BL/10ScSnJ) or *mdx* (C57BL/10ScSn-Dmd < *mdx* > /J) mouse pups (P1 or P2) were sacrificed by cervical dislocation. Hearts were rapidly removed and placed in falcon tubes with ice-cold ADS buffer consisting of 106 mM NaCl, 20 mM Hepes, 0.8 mM NaH_2_PO_4,_ 0.4 mM MgSO4, and 5 mM glucose. The ventricles were extracted and minced into small 1–3 mm pieces and digested with a solution containing 0.45 mg/mL Collagenase (Roche, Cat# 10103586001, Rotkreuz, Switzerland) and 1.25mg/mL Pancreatin (Sigma-Aldrich, Cat# P1750 25 mg, Milwaukee, WI, USA). Digestions steps were carried out by incubating the tissues with 9 mL Collagenase solution at 37 °C and constantly shaking. After 10 min the supernatant was collected and neutralized with 1 mL of Newborn Calf Serum (NCS) (Thermo Fisher Scientific, 16010159, Waltham, MA, USA). Cells were centrifuged and resuspended in 2 mL of Newborn Calf Serum (NCS). 3 to 4 rounds of the same digestion step were carried out to dissociate all the tissue. After digestion, the collected cells were pooled together and centrifuged at 1250 rpm for 5 min. The cell pellet was then resuspended in culture medium 1 (M1) consisting of 65.3% DMEM 25 mM Hepes (Thermo Fisher Scientific, Cat# 42430025, Waltham, MA, USA), 17.07% M-199 (Thermo Fisher Scientific, Cat# 31150022, Waltham, MA, USA), 9.75% Horse Serum (Thermo Fisher Scientific, 26050070, Waltham, MA, USA), 4.87% Newborn Calf Serum (Thermo Fisher Scientific, 16010159, Waltham, MA, USA), 0.975% 200 mM L-Glutamine (Thermo Fisher Scientific, 25030149, Waltham, MA, USA), 0.975% Insulin-Transferrin-Selenium-Supplement 100X (Thermo Fisher Scientific, Cat# 41400045, Waltham, MA, USA), and 0.0975% Penicillin-Streptomycin (Thermo Fisher Scientific, 15140148, Waltham, MA, USA) (all percentages vol/vol). Cells were seeded onto 10 cm cell-culture dishes and incubated for 2 h at 37 °C to allow attachment of fibroblasts to the plastic. After 2 h the supernatant, containing an enriched population of cardiomyocytes, was collected and centrifuged. The cardiomyocyte pellet was then resuspended in medium M1 and the cells seeded on laminin pre-coated 15 mm diameter sterilised borosilicate glass coverslips. The coverslips were coated at least 1 h before seeding with a solution of 100 μg/mL Laminin (mouse) (BD Biosciences, Cat# 354232, San Jose, CA, USA) diluted 1:5 in phosphate-buffered saline solution. At day 2 the wells were washed 3× with ADS buffer and culture medium 2 (M2), consisting of 74.62% DMEM 25 mM Hepes, 16.9% M-199, 4.97% Horse Serum, 0.49% Newborn Calf Serum, 0.991% 200 mM Glutamine, 0.99% Insulin-Transferrin-Selenium-Supplement 100X, and 0.099% of Penicillin-Streptomycin (all percentages vol/vol), was added to the wells. Cells were transduced after 2 h of incubation at 37 °C.

### 4.4. Viral Infection

Cardiac myocytes were transduced with second-generation adenoviral vectors encoding for the different FRET-based sensors at day 2 in culture. A volume of adenoviral vector suspension corresponding to multiplicity of infection (MOI) of 30 was added to each well. After 2 h incubation at 37 °C the medium containing the virus was removed and replaced with fresh culture medium M2. Cells were placed back in the incubator and imaging was carried out after 12 to 24 h.

### 4.5. FRET Imaging

FRET imaging was performed on an inverted microscope (Olympus IX71, Olympus, Southend-on-Sea, UK) using a PlanApoN, 60×, NA 1.42 oil immersion objective, 0.17/FN 26.5 (Olympus, Southend-on-Sea, UK). The microscope was equipped with a coolSNAP HQ^2^ monochrome camera (Photometrics, Tucson, AZ, USA) and an optical beam-splitter device required for the simultaneous recording of the YFP and CFP emissions (Dual-view simultaneous-imaging system, DV2 MAG Biosystems, Photometrics, Tucson, AZ, USA). The FRET filter settings used throughout were: CFP excitation filter ET436/20x, dichroic mirror 455DCLP (Chroma Technology, Maximilianstraße 33, 82140 Olching, Germany) in the microscope filter cube; dichroic mirror 505DCLP, YFP emission filter 545 nm, and CFP emission filter 480 nm (Chroma Technology, Olching, Germany) in the beam splitter. Images were acquired and processed using Meta imaging series 7.1, MetaFluor, (Molecular Devices, San Jose, CA, USA). Fluorescence emission images were acquired every 10 s. Basal FRET was measured as the background-subtracted 480 nm/545 nm fluorescence emission intensity on excitation at 436 nm. FRET changes were measured as changes in the background-subtracted 480 nm/545 nm fluorescence emission intensity on excitation at 430 nm and expressed as R/R0, where R is the 480 nm/545 nm value at time t and R0 is the average ratio of the 8 frames preceding addition of the stimulus. Cells were cultured on 15 mm diameter sterilised borosilicate glass coverslips, which allow the transfer to a perfusion chamber. The coverslips were held in place with a thin rim of silicone vacuum grease (18405-Sigma-Aldrich, Milwaukee, WI, USA) in order to create a water-tight seal between the chamber and the coverslip. The chamber allowed the application of a stimulus to the bath of superfusate.

During the experiment cells were maintained at room temperature in a solution consisting of 125 mM NaCl, 5 mM KCl, 1 mM Na_3_PO_4,_ 1 mM MgSO_4,_ 20 mM Hepes, 5.5 mM Glucose and 1 mM CaCl_2._ Reagents used on experiments: Isoproterenol-hydrochloride (Sigma-Aldrich Milwaukee, WI, USA), Forskolin (Sigma-Aldrich, Milwaukee, WI, USA), IBMX (Sigma-Aldrich, Milwaukee, WI, USA), Bay 60-7550 (Cayman Chemicals, Neratovice, Czech Republic), Cilostamide (Cayman Chemicals, Ann Arbor, MI, USA), Rolipram (Cayman Chemicals, Ann Arbor, MI, USA), SNAP (Cayman Chemicals, Ann Arbor, MI, USA), Bay 41-2272 (Sigma-Aldrich, Milwaukee, WI, USA), Sildenafil-citrate (Sigma-Aldrich, Milwaukee, WI, USA), PF-04447943 (Sigma-Aldrich, Milwaukee, WI, USA), PF-04957325 (gift from Pfizer, now available from MedChemExpress, USA), Cinaciguat-hydrochloride (Sigma-Aldrich, Milwaukee, WI, USA).

### 4.6. Confocal Imaging

Twenty-four hours after infection, the cardiomyocytes were incubated with the membrane dye Wheat Germ Agglutinin (WGA) Alexa Fluor 594 Conjugated (5 µg/mL) (Thermo Fisher Scientific, Cat# W11262, Waltham, MA, USA) for 10 min or with MitoTracker red (Thermo Fisher Scientific, Cat# M7512, Waltham, MA, USA) 1:10.0000 for 5 min at 37 °C, for co-localization imaging of the plasmalemma or the OMM, respectively. Stained cells were then fixed with 4% formaldehyde and rinsed with PBS + 0.1% Tween 20. For imaging, coverslips were mounted onto a microscope slide with Ibidi Mounting Medium (Ibidi, Cat# 50001, Martinsried, Germany). Images were acquired with an Inverted Olympus FV1000 confocal microscope and processed with the image processing program ImageJ [59]. Colocalization of sensors and markers (WGA and MitoTracker) was assessed by calculating the Pearson’s correlation Coefficient by using the ImageJ plugin Coloc2.

### 4.7. qPCR

RNA was extracted from isolated cardiomyocytes using TRIzol Reagent (Thermo Fisher Scientific, Cat# 15596026, Waltham, MA, USA) following manufacturer’s instructions. RNA was quantified using Nanodrop. Reverse transcription of total RNA was performed by using High-Capacity cDNA Reverse Transcription Kit (Applied Biosystems/Thermo Fisher Scientific, Cat# 4368814, Waltham, MA, USA) following manufacturer’s instructions. 1 µM RNA was used in the reaction. 1/5 dilution of cDNA (reaching 25 ng) was used in each qPCR reaction. Samples were plated in triplicate and qPCR reaction was performed by using Taqman probes for PDE1A (Mm00450244_m1), PDE1C (Mm00478051_m1), PDE2A (Mm01136644_m1), PDE3A (Mm00479581_m1), PDE3B (Mm00691635_m1), PDE4A (Mm01147149_m1), PDE4B (Mm00480174_m1), PDE4D (Mm00456879_m1), PDE5A (Mm00463177_m1), PDE8A (Mm01315378_m1), PDE9 (Mm00501039_m1), AC5 (Mm00674122_m1) and AC6 (Mm00475772_m1).

### 4.8. Western Blotting

Protein was extracted from isolated neonatal cardiomyocytes using RIPA buffer (Sigma-Aldrich, Milwaukee, WI, USA) supplemented with cOmplete Mini Protease Inhibitor Cocktail (Roche, Rotkreuz, Switzerland). Total protein quantification was determined using BCA protein assay (Sigma-Aldrich, Milwaukee, WI, USA). 40µg of total control or MDX protein were loaded into NuPAGE 4–12% Bis-Tris Gel (Invitrogen/Thermo Fisher Scientific, Waltham, MA, USA). The membrane was then incubated with antibody against the β1-subunit of sGC (GUCY1B3), from Abcam (ab154841) at a 1:250 dilution and the GAPDH antibody from Proteintech (60004-1-Ig) at 1:1000 dilution.

### 4.9. Statistics

Data are presented as means ± standard error of the mean. Graph Pad Prism 5 software (version 5 for windows, GraphPad Software, La Jolla, CA, USA) was used for all statistical analyses. Student *t*-test was used to compare two groups of experiments. The comparison of 3 or more groups was performed using ANOVA test with Bonferroni’s correction. The significance level was set at *p* < 0.05 and the following annotation has been used: * 0.01 ≤ *p* ≤ 0.05, ** 0.001 ≤ *p* < 0.01, *** *p* < 0.001. Number of biological replicates *n* are stated in the figure legends. Biological replicates are independent cultures, each of which was generated from 10–24 pups.

## Figures and Tables

**Figure 1 ijms-21-07056-f001:**
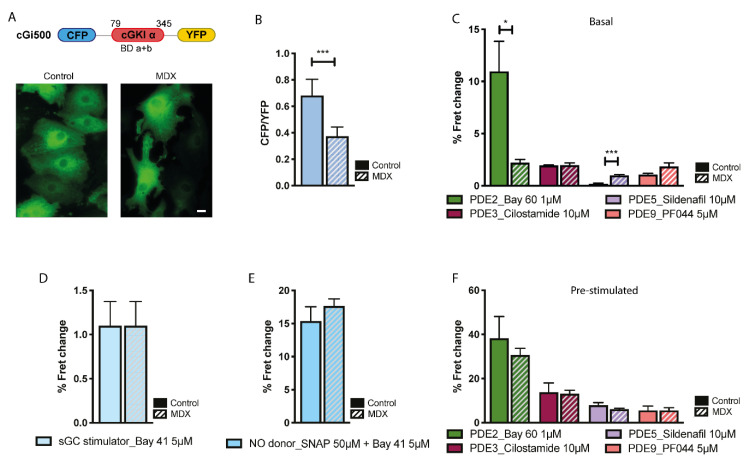
Comparison of cGMP signals in the bulk cytosol of neonatal ventricular myocytes (NVM) from control and *mdx* mice. (**A**) Top: schematic representation of the cytosolic cGMP fluorescence resonance energy transfer (FRET) reporter cGi500. Bottom: representative image of control and *mdx* NVM expressing cGi500. (**B**) Mean relative values for CFP to YFP fluorescence intensity ratio recorded in unstimulated NVM from control and *mdx* mice expressing cGi500. *n* = 4. (**C**) Mean FRET change recorded in NVM on application of family-selective cGMP-PDE inhibitors as indicated. *n* = 4. (**D**) Mean FRET change recorded in NVM from control or *mdx* mice on application of the cGC stimulator Bay-41. *n* = 2. (**E**) Mean FRET change recorded in NVM from control or *mdx* mice on application the NO donor SNAP in combination with the cGC stimulator Bay-41. *n* = 4. (**F**) Mean FRET change recorded in NVM on application of family-selective cGMP-PDE inhibitors as indicated in cells pre-treated as in (**E**) *n* = 4. For all data, mean ± SEM are shown. * = 0.01 ≤ *p* ≤ 0.05, *** = *p* < 0.001. Size bar: 10 µm.

**Figure 2 ijms-21-07056-f002:**
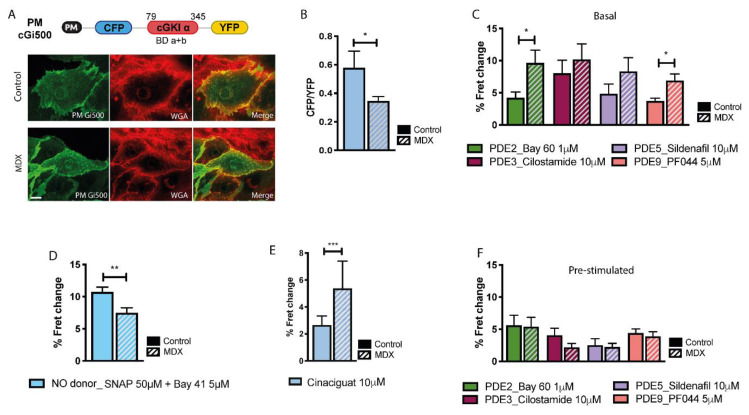
Comparison of cGMP signals at the plasmalemma of NVM from control and *mdx* mice. (**A**) Top: schematic representation of the plasmalemma-targeted cGMP FRET reporter PM-cGi500. PM, plasma-membrane targeting domain (see main text for details); cGKIα indicates cGMP-dependent protein kinase. Numbers on top indicate amino acid position (bovine sequence). BD a + b indicates cGMP binding domain (a and b) from cGKI. Bottom: representative image of control and *mdx* NVM expressing PM-cGi500 (left). Co-staining with the plasma membrane dye wheat germ agglutinin (WGA) is shown in the central panels and the overlay is shown at the right. (**B**) Mean relative values for CFP to YFP fluorescence intensity ratio recorded in unstimulated NVM from control and *mdx* mice expressing PM-cGi500. *n* = 4. (**C**) Mean FRET change recorded in NVM expressing PM-cGi500 on application of family-selective cGMP-PDE inhibitors as indicated. *n* = 4. (**D**) Mean FRET change recorded in NVM from control or *mdx* mice expressing PM-cGi500 on application of the NO donor SNAP in combination with the cGC stimulator Bay-41. *n* = 4. (**E**) Mean FRET change recorded in NVM from control or *mdx* mice expressing the PM-cGi500 sensor upon addition of 10 μM cinaciguat. *n* = 3. (**F**) Mean FRET change recorded in NVM expressing PM-cGi500 on application of family-selective cGMP-PDE inhibitors, as indicated, in cells pre-treated as in (**D**). *n* = 4. For all data, mean ± SEM are shown. * = 0.01 ≤ *p* ≤ 0.05, ** = 0.001 ≤ *p* < 0.01, *** = *p* < 0.001. Size bar: 10 µm.

**Figure 3 ijms-21-07056-f003:**
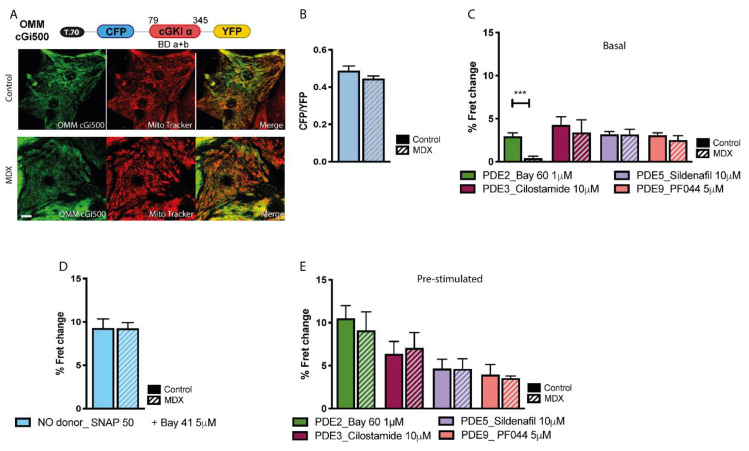
Comparison of cGMP signals at the outer mitochondrial membrane (OMM) of NVM from control and *mdx* mice. (**A**) Top: schematic representation of the OMM-targeted cGMP FRET reporter OMM-cGi500. T.70 indicates translocase outer mitochondria membrane protein TOM 70. Bottom: at the left, representative images of NVM from control and *mdx* mice expressing OMM-cGi500. Co-staining with the mitochondria specific dye mitotracker is shown in the central panel and the overlay is shown at the right. (**B**) Mean relative values for CFP to YFP fluorescence intensity ratio recorded in unstimulated NVM from control and *mdx* mice expressing OMM-cGi500. *n* = 4. (**C**) Mean FRET change recorded in NVM expressing OMM-cGi500 on application of family-selective cGMP-PDE inhibitors as indicated *n* = 4. (**D**) Mean FRET change recorded in NVM from control or *mdx* mice expressing OMM-cGi500 on application of the NO donor SNAP in combination with the cGC stimulator Bay-41. *n* = 4. (**E**) Mean FRET change recorded in NVM expressing OMM-cGi500 on application of family-selective cGMP-PDE inhibitors, as indicated, in cells pre-treated as in (**D**) *n* = 4. For all data, mean ± SEM are shown. *** = *p* < 0.001. Size bar: 10 µm.

**Figure 4 ijms-21-07056-f004:**
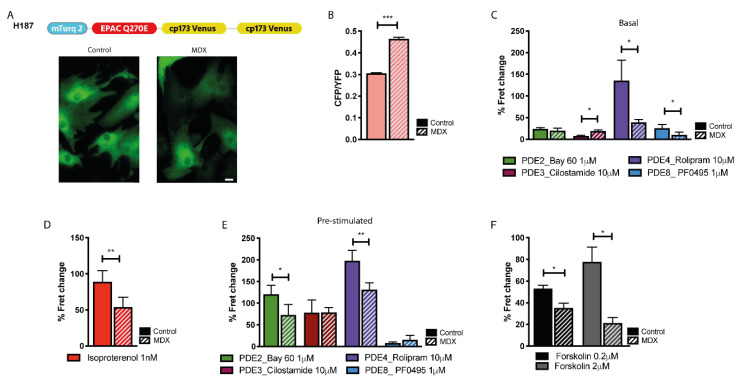
Comparison of cAMP signals in the cytosol of NVM from control and *mdx* mice. (**A**) Top: schematic representation of the cytosolic cAMP FRET reporter H187. Bottom: representative images of control and *mdx* NVM expressing H187. (**B**) Mean relative values for CFP to YFP fluorescence intensity ratio recorded in unstimulated NVM from control and *mdx* mice expressing H187. *n* = 4 (**C**) Mean FRET change recorded in NVM expressing H187 on application of family-selective cAMP-PDE inhibitors as indicated. *n* = 4. (**D**) Mean FRET change recorded in NVM from control or *mdx* mice expressing H187 on application of ISO, as indicated. *n* = 4. (**E**) Mean FRET change recorded in NVM expressing H187 on application of family-selective cAMP-PDE inhibitors, as indicated, in cells pre-treated with ISO 1 nM. *n* = 4. (**F**) Mean FRET change recorded in NVM from control and *mdx* mice expressing H187 on application of the AC activator forskolin, as indicated. *n* = 3. For all data, mean ± SEM are shown. * = 0.01 ≤ *p* ≤ 0.05, ** = 0.001 ≤ *p* < 0.01, *** = *p* < 0.001. Size bar: 10 µm.

**Figure 5 ijms-21-07056-f005:**
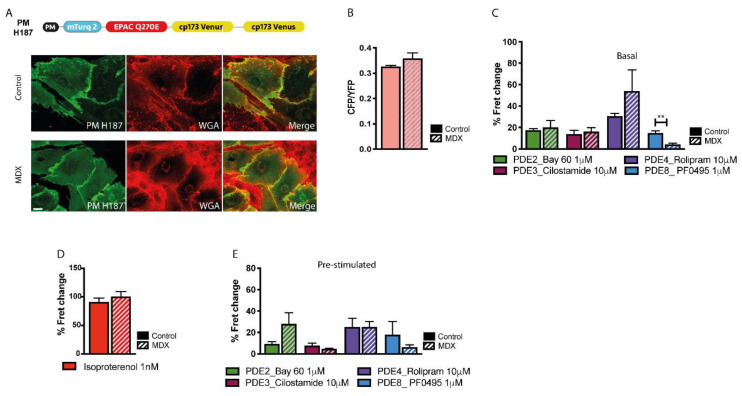
Comparison of cAMP signals at the plasmalemma of NVM from control and *mdx* mice. (**A**) Top: schematic representation of the plasmalemma-targeted cAMP FRET reporter PM-H187. PM, plasma-membrane targeting domain. Bottom: left, representative images of control and *mdx* NVM expressing PM-H187. Co-staining with the plasma membrane dye WGA is shown in the central panel and the overlay is shown on the right. (**B**) Mean relative values for CFP to YFP fluorescence intensity ratio recorded in unstimulated NVM from control and *mdx* mice expressing PM-H187. *n* = 4. (**C**) Mean FRET change recorded in NVM expressing PM-H187 on application of family-selective cAMP-PDE inhibitors as indicated. *n* = 4. (**D**) Mean FRET change recorded in NVM from control or *mdx* mice expressing PM-H187 on application of ISO 1 nM. *n* = 4. (**E**) Mean FRET change recorded in NVM expressing PM-H187 on application of family-selective cAMP-PDE inhibitors, as indicated, in cells pre-treated with ISO 1 nM. *n* = 4. For all data, mean ± SEM are shown. ** = 0.001 ≤ *p* < 0.01. Size bar: 10 µm.

**Figure 6 ijms-21-07056-f006:**
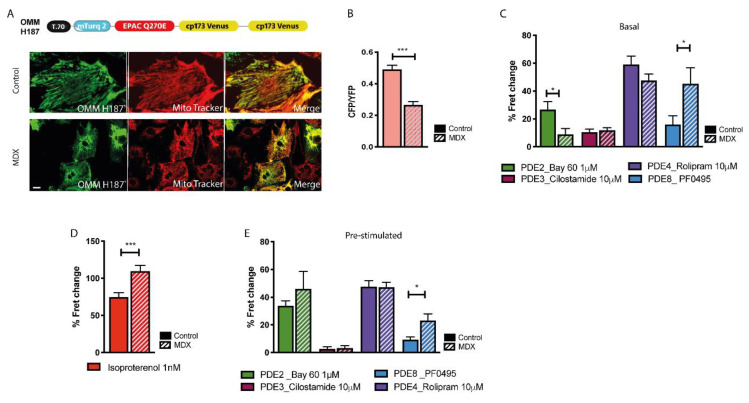
Comparison of cAMP signals at the OMM of NVM from control and *mdx* mice. (**A**) Top: schematic representation of the OMM-targeted cAMP FRET reporter OMM-H187. Bottom: left, representative images of control and *mdx* NVM expressing OMM-H187. Co-staining with the mitochondria specific dye mitotracker is shown in the central panels and the overlay is shown on the right. (**B**) Mean relative values for CFP to YFP fluorescence intensity ratio recorded in unstimulated NVM from control and *mdx* mice expressing OMM-H187. *n* = 4. (**C**) Mean FRET change recorded in NVM expressing OMM-H187 on application of family-selective cAMP-PDE inhibitors as indicated. *n* = 4. (**D**) Mean FRET change recorded in NVM from control or *mdx* mice expressing OMM-H187 on application of ISO 1 nM. *n* = 4. (**E**) Mean FRET change recorded in NVM expressing OMM-H187 on application of family-selective cAMP-PDE inhibitors, as indicated, in cells pre-treated with 1 nM ISO. *n* = 4. For all data, mean ± SEM are shown. * = 0.01 ≤ *p* ≤ 0.05, *** = *p* < 0.001. Size bar: 10 µm.

**Figure 7 ijms-21-07056-f007:**
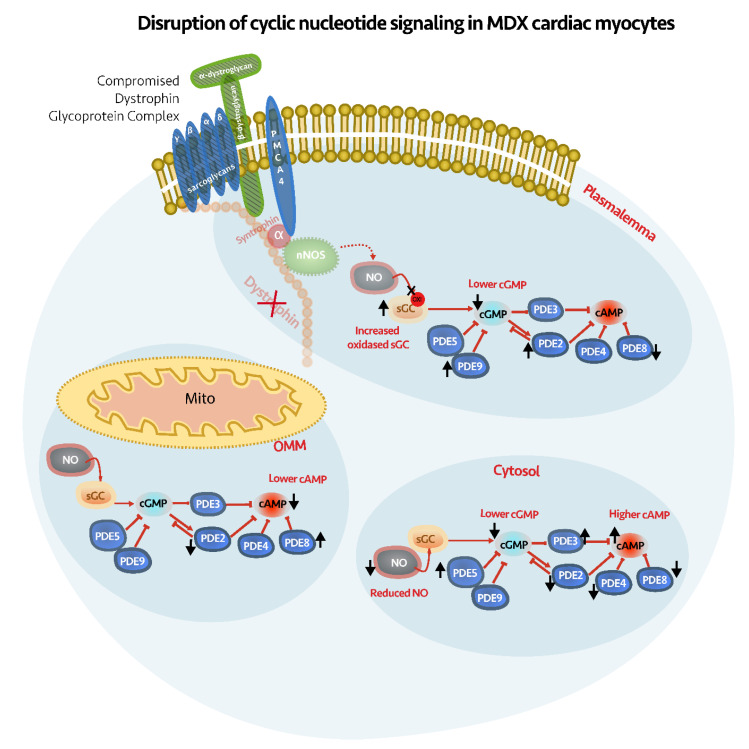
Schematic illustration of how basal cGMP and cAMP levels are affected at the plasmalemma, OMM and cytosol of NVM from *mdx* mice. Black arrows indicate changes that occur in the activity of PDEs, NO levels or amount of oxidised sGC in the dystrophic cells compared to controls.

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
