# Peer review of "Multi-Compartment, Early Disruption of cGMP and cAMP Signalling in Cardiac Myocytes from the mdx Model of Duchenne Muscular Dystrophy"

_ijms, 2020, doi:10.3390/ijms21197056_

Round 1

Reviewer 1 Report

The aim of this paper  to clarify whether compartmentalized cyclic nucleotide signals are altered in Duchenne muscular dystrophy (DMD). In that way, the authors by using FRET imaging analyzed on one side the participation of cGMP hydrolyzing PDEs: PDE2, PDE3, PDE5 and PDE9 and  on the other side cAMP hydrolyzing enzymes: PDE2, PDE3, PDE4 and PDE8. This study was done in different compartment cells; cytosol, plasmalemma and outer mitochondria membrane of control and mdx cardiac myocytes, hoping to clarify relationships between DMD, and PDEs compartmentations. This interesting study focused on different PDEs, however this study cannot exclude the contributions of PDE1s in the different studied compartments  and more over the nucleus compartment in DMD. Since FRET imaging might be limited by its technology and by the choice of used inhibitor concentrations, another technical approach such as  Western blot (possibly PDE assays) is recommended to analyze and integrate the obtain data, allowing to conceive a final schema showing the complex reorganization of PDE activities.

1) PDE isozymes have been studied at the mRNA level which might not reflect the changes of corresponding protein levels which might be of interest in this study.  It would be better in the supplementary Figure 1 to know the relative participation of each PDEs where  mRNA levels were missing for PDE4C, PDE8B and  PDE1A, 1B, 1C, instead of  their changes, the same concern for supplementary Figure 2.

2) To assess the changes in PDE activities analyzed by fret, it is necessary to performed Western blot  such as previously done (Monterisi, et al.,2017) in control and MDX cardiomyocytes for the investigated PDEs and also for PDE4C (although it is not detected in normal tissue), PDE8B and  PDE1A, 1B, 1C  (interestingly PDE protein expression  of PDE4A, PDE4D, PDE5A and PDE1s changed at the cardiac level: Mokni et al., 2010). Of interest, Western blots  might allow to quantify the distribution of each PDE in the whole different (outer and inner MM)  isolated cellular compartments to confirm and develop previous FRET data.

3) It is important to know how the choice of concentration inhibitors used in  this FRET study  was performed to assess the participation of each PDEs. Notably,  justify the use of 10 µM sildenafil to inhibit PDE5, as much as this concentration might also inhibits PDE1 and PDE9 (see Corbin et al., UROLOGY2002, 60 (Suppl 2B): 4–11). Therefore 10 µM of sildenafil might inhibit not only PDE5, but also PDE1 and PDE9, inducing miss-interpretations.

4) Concerning ref 41, it was recently reported (Tetsi et al., 2019) that sildenafil tends to reduce mitochondrial ROS production in ischemic skeletal muscle.

5) It would be necessary to add another revue concerning kinetic properties of PDE8 and PDE4 such as Keravis, Lugnier, 2012.

6) Since, the authors suggest “a redistribution of PDE8 between compartments in dystrophic myocytes, to assess this hypothesis it is necessary to performed Western blots in the different subcellular compartments or confocal studies.

Minor:

 Along the manuscript, it is necessary to replace in the figures and in their legends: BL 10 by Control.

 Legend of Figure 5B is missing.

Author Response

Reviewer #1

The aim of this paper  to clarify whether compartmentalized cyclic nucleotide signals are altered in Duchenne muscular dystrophy (DMD). In that way, the authors by using FRET imaging analyzed on one side the participation of cGMP hydrolyzing PDEs: PDE2, PDE3, PDE5 and PDE9 and  on the other side cAMP hydrolyzing enzymes: PDE2, PDE3, PDE4 and PDE8. This study was done in different compartment cells; cytosol, plasmalemma and outer mitochondria membrane of control and mdx cardiac myocytes, hoping to clarify relationships between DMD, and PDEs compartmentations. This interesting study focused on different PDEs, however this study cannot exclude the contributions of PDE1s in the different studied compartments  and more over the nucleus compartment in DMD. Since FRET imaging might be limited by its technology and by the choice of used inhibitor concentrations, another technical approach such as  Western blot (possibly PDE assays) is recommended to analyze and integrate the obtain data, allowing to conceive a final schema showing the complex reorganization of PDE activities.

We thank this reviewer for the positive comments on the manuscript and the suggestions for further investigations. We would like to remark, however, that the focus of our study was not ‘to clarify relationships between DMD, and PDEs compartmentations’ but to assess whether cyclic nucleotides (CNs) signalling is altered in neonatal ventricular myocytes from mdx hearts compared to healthy controls. In our experiments, we used PDE inhibitors as a way of perturbing locally CNs levels. Although we believe our results suggest a different contribution of specific PDEs to local CNs signals, we did not intend to make definitive claims with respect to the different PDE isoforms present in the compartments we investigated. We apologise if the aim of the study was not clearly presented and we have made efforts in the revised manuscript to clarify this point (see for example end of first paragraph in Results). With respect to CNs signals in the nucleus, indeed we did not explore this compartment. Similarly, we did not look at other subcellular sites that may be of interest (e.g. the endoplasmic reticulum, the myofilaments etc). Again, our aim in this study was not to present a comprehensive analysis of subcellular compartments and the focus was on the PM, the OMM and the bulk cytosol, sites that we considered could be directly affected by the lack of dystrophin.

1) PDE isozymes have been studied at the mRNA level which might not reflect the changes of corresponding protein levels which might be of interest in this study.  It would be better in the supplementary Figure 1 to know the relative participation of each PDEs where mRNA levels were missing for PDE4C, PDE8B and  PDE1A, 1B, 1C, instead of  their changes, the same concern for supplementary Figure 2.

We agree with this reviewer that to characterize the protein levels for the different PDE isoforms may be of interest. However, as discussed above, this was not the focus of our study as here we were primarily concerned with the characterization of the CNs response in different subcellular compartments, as we have tried to reflect in the wording of title, introduction, results and discussion. In our experiments, we used PDE family-selective inhibitors as a way of manipulating CNs levels to ascertain whether there are differences in the compartments under investigation. Although a full characterization of individual PDE isoforms in different subcellular compartments would be of interest, this is an extremely taxing endeavor, given the number of isoforms, particularly within some PDE families, and the necessity to validate the specificity of PDE isoform-specific antibodies. We posit that, although of interest, this should be the focus of a separate study.

On the other hand, we took on board the point raised by this reviewer that some of the PDE isoforms were missing in our original mRNA characterization. Unfortunately, due to COVID-19 associated restrictions imposed by our animal facility, we are currently unable to perform animal work and have not been able to generate fresh cultures from which to extract mRNA. Having to rely on a limited supply of stored mRNA, we could not perform the qPCR experiment for all PDE isoforms simultaneously but had to limit the analysis of additional isoforms to PDE1A, PDE1C and PDE8A. We now show this data in the revised Supplementary Fig 4, were the level of mRNA is shown for each isoform relative to the other isoforms. Isorom exclusion was guided by the fact that PDE1B is not detected in mouse heart (doi: 10.1073/pnas.89.22.11079) and PDE8B is present in lower abundance  than PDE8A in mouse heart (doi:10.1016/j.yjmcc.2010.03.016). We also revised Supplementary fig 6 where the AC data are presented as relative to each other.

2) To assess the changes in PDE activities analyzed by fret, it is necessary to performed Western blot  such as previously done (Monterisi, et al.,2017) in control and MDX cardiomyocytes for the investigated PDEs and also for PDE4C (although it is not detected in normal tissue), PDE8B and  PDE1A, 1B, 1C  (interestingly PDE protein expression  of PDE4A, PDE4D, PDE5A and PDE1s changed at the cardiac level: Mokni et al., 2010). Of interest, Western blots might allow to quantify the distribution of each PDE in the whole different (outer and inner MM)  isolated cellular compartments to confirm and develop previous FRET data.

 As articulated above, although we agree that a full characterization of the individual PDE isoforms that operate in subcellular compartments would be interesting, this was not the focus of our study. Here, we sought to establish whether there are differences in the cAMP and cGMP signals in subcellular compartments of mdx vs healthy myocytes. In this respect, we would like to point out that the presence of a certain combination of PDEs at a given subcellular site does not, per se, determine the level of CNs at that location, as the regulatory state and consequent level of activity of the local PDEs will have a major impact. Therefore, establishing the amount of PDE protein at a certain site provides a different type of information than the information we can obtain by directly measuring, with the targeted  reporters, the level of second messenger achieved on inhibition of a particular PDE family. In addition, measuring PDE activity using a conventional PDE-activity assay, would not provide information on the contribution of PDEs to local levels of CNs as those assays are performed on cell lysates with complete loss of any subcellular resolution.

3) It is important to know how the choice of concentration inhibitors used in this FRET study was performed to assess the participation of each PDEs. Notably,  justify the use of 10 µM sildenafil to inhibit PDE5, as much as this concentration might also inhibits PDE1 and PDE9 (see Corbin et al., UROLOGY2002, 60 (Suppl 2B): 4–11). Therefore 10 µM of sildenafil might inhibit not only PDE5, but also PDE1 and PDE9, inducing miss-interpretations.

We are aware that 10 μM sildenafil partly inhibits also PDE1 and PDE9. However, we would argue that this is not a limitation, as the aim of our study was not to establish the contribution of different PDEs but to establish whether there are differences between control and mdx in terms of local CNs signals. We realise, however, that the incomplete selectivity of sildenafil at the concentration used should be explicitly acknowledged and we have added a comment to this effect in the revised text on p 10, middle paragraph.   

4) Concerning ref 41, it was recently reported (Tetsi et al., 2019) that sildenafil tends to reduce mitochondrial ROS production in ischemic skeletal muscle.

 This reference is now being included in the revised manuscript on p 13, last paragraph

5) It would be necessary to add another revue concerning kinetic properties of PDE8 and PDE4 such as Keravis, Lugnier, 2012.

The suggested review has been included on page 15, top

6) Since, the authors suggest “a redistribution of PDE8 between compartments in dystrophic myocytes, to assess this hypothesis it is necessary to performed Western blots in the different subcellular compartments or confocal studies.

We agree with this reviewer that, given our FRET results, it would be interesting to establish whether there is indeed a redistribution of PDE between compartments in mdx cells. We did attempt this experiment with the limited amount of tissue we had available but we were not able to detect any signal. This may be due to partial degradation of the protein in our stored samples. As mentioned above, due to COVID-19 related restrictions imposed by our animal facility, we are not currently able to generate fresh samples. We therefore rephrased our statement on p 15, end of first paragraph, to stress the fact that the redistribution of PDE8 between compartments is currently an hypothesis that will need to be confirmed experimentally.

Minor:

 Along the manuscript, it is necessary to replace in the figures and in their legends: BL 10 by Control.

Amended

Legend of Figure 5B is missing.

The legend to Figure 5B is now included

Reviewer 2 Report

The manuscript of Brescia et al. describes the development of three novel targeted sensors for the cyclic nucleotides cGMP and cAMP (PM-cGi500, OMM-cGi500 and PM-H187). These, in addition to three previously described sensors were expressed in neonatal left ventricular cardiomyocytes from C57BL mice. The sensors detected cGMP or cAMP in the bulk cytosol, at the plasma membrane and at the outer mitochondrial membrane in the basal and cell-stimulated state. In addition to this, the activity of several phosphodiesterases (PDEs) in the different subcellular compartments were analyzed in the basal and cell-stimulated state. The findings in normal mice were compared to a model of Duchenne Muscular Dystrophy termed mdx mice. Duchenne is often a result of functional loss of dystrophin that disrupts in sarcolemma integrity, and in cardiomyocytes to disruption of Ca2+ homeostasis, decreased NO production and mitochondrial signaling.

In the mdx model, the authors show that basal cGMP levels are decreased, specifically around the plasma membrane and that this might be a result of increased local PDE activity of certain isoforms. Beta-adrenergic receptor activation increases cAMP levels and reveals a different distribution of PDEs in the mdx model. These data are nicely modeled to a two-compartment model showing how the complex distribution of PDEs modulate cAMP levels at the outer mitochondrial membrane.

The manuscript is well written and contains detailed information on how soluble GC and beta-adrenergic receptor signaling is modified in the mdx model. The manuscript provides significant advances in tools for localized signaling of cyclic nucleotides and also significant advances in understanding the localization of both stimulators and PDEs in the plasma membrane and outer mitochondrial membrane of cardiomyocytes.

Major concerns:

The authors need to better describe the rationale for choosing to construct plasmamembrane-localized sensors. There is a larger rationale to do this for the cAMP-sensor, since beta adrenergic signaling occurs at the plasma membrane, but it is less clear to this reviewer that soluble GC is restricted to the plasma membrane (about 2/3 is present in the cytosol DOI: 10.1161/CIRCRESAHA.111.259242). Please provide a rationale for a PM-localized cGMP sensor. Determining cGMP at the plasma membrane from stimulation with natriuretic peptides would provide a better localized response than from sGC. Please provide information or data showing localization of sGC, and also if this localization is changed in the MDX model. This is particularly important to determine if the reduced sGC seen in the PM sensor (and not in the cGi500 or OMM sensors) is due to altered localization of sGC.

Image quality: The fluorescent images in the manuscript version sent from the journal to this reviewer was of poor quality with a high amount of distortion. I suspect that this is due to technical difficulties in the manuscript sent to the reviewers and that the printed version is of higher quality.

Sensor localization: As judged by the images provided, the authors need to determine if there is a nuclear localization of PM-cGi500, PM-H187 and OMM-H187? I noticed that the authors included DAPI in their methods section but not in their fluorescent images. Could DAPI be included to judge nuclear localization of these sensors? If there is a high degree of nuclear localization, was this included in the region of interest analyzed for FRET? In addition, possibly due to the reduced quality of the images received, is there significant overlap between OMM-cGi500 and the mitotracker used? A Pearson correlation coefficient (or similar) would be helpful to judge the co-localization.

Since the MDX model alters localization of signalosomes, is the localization of the localized sensors preserved in the MDX model? This is particularly important for the OMM sensors, since the authors refer to mitochondrial biogenesis being altered in DMD.

Traces: There is a lack of representative FRET traces to show the behavior of these newly developed sensors. Please provide such representative traces.

Saturation and dynamic range of localized sensors: In some of the sensors, such as the PM-cGi500, there appears to be a lower FRET change in the presence of stimulator than in the absence (e.g. Bay60 and PF044 for MDX). There could be several reasons for such a discrepancy, including different source of the basal cGMP compared to the sGC-stimulated or it could be due to saturation of the sensor and therefore not achieving higher FRET changes. Determining this is important, since the authors have previously shown that some sensors yield a decreased efficacy in the presence of a targeting domain (DOI:10.1038/ncomms15031). Could the authors provide data that the localized sensor are saturated after for example sGC-stimulated and PDE2-inhibited cells in the MDX model (PM-cGi500) by subsequent addition of either IBMX or natriuretic peptides? The H187 sensor seems to provide much lower FRET changes in the targeted sensors upon PDE-Inhibition despite similar increase in isoproterenol-stimulated FRET changes. Is this due to reduced efficacy of the localized sensors (could be determined with addition of forskolin) or reduced localization of PDEs in either the PM or OMM?

Selectivity of sildenafil: The IC50 of sildenafil for PDE5 was found to be ~7 nM, whereas the IC50 for PDE1 was ~200 nM (DOI: 10.1124/jpet.109.154468.). Since the authors uses ~1000 the IC50 of PDE5 (10 µM), which would also inhibit PDE1, does this change the interpretation of involvement of PDE5 vs. PDE1 in the PM-cGi500 sensor measurements? This would be particularly important if there is an increase in expression of PDE1 in the MDX model (which was not analyzed in Supplementary Figure 1).

The authors have performed a large amount of work to include several variables (two models, PDE-inhibition in basal and stimulated cells) that creates several permutations. Are all the cells included in the measurements of basal FRET and stimulated FRET? If not, is there high enough statistical power to determine some of these differences? Depending on whether there are 4 cells, an average of 4 biological repeats or even 16 cells from 4 biological repeats, the statistical analysis of changes in Figure 4D should be performed again, as the error bars are greatly overlapping, while a p<0.01 is stated.

In some of the experiments, a higher basal FRET was coinciding with a larger PDE activity of some PDEs (eg. PDE2 in cGi500, PM-cGi500 and OMM-H187, PDE9 in PM-cGi500 and PDE8 in OMM-H187). Did I miss a comment on this in the discussion section?

The abstract does not contain a summary of your findings. The authors must indicate their findings in the abstract.

Illustration: It would be very helpful for the reader with an illustration of the localization of different PDEs and how these are altered in the MDX model.

Minor:

Figure legends: There is a large amount of information missing in figure legend 1E and 5B.

Figure caption: In PDE experiments, it might be difficult for the reader to understand the difference between inhibiting PDEs in the absence (basal) and presence of pre-stimulation without reading the figure legend. Please indicate on the panels if it is basal or pre-stimulation of a cAMP/cGMP elevating drug.

Error bars: Please keep error bars uniform. In some panels (e.g. Figure 1C), there is a mix between upward and error bars in both directions.

The authors did not see much cGMP using the sGC stimulator and performed all experiments in the presence of the NO donor SNAP in addition to sGC stimulator. Although the sGC stimulator enhances NO-stimulated sGC, I wondered whether the authors compared SNAP alone with SNAP + Bay41?

Supplementary Figure 3: To aid the reading and understanding of this figure, it might be useful with a heading for each panel. In line with this, the heading “Perturbed model” seems difficult to understand. Since this model assumes that there is no alteration of PDE8 activity in the MDX model, altering the heading to include this might be useful.

Reference list: please include all authors or insert “et al.” after the 10 authors you have listed according to the journal’s guidelines. Please update formatting of reference 41.

Line 52: 2+ in superscript

Line 90: insert a period after “isolation”.

Line 97-98: please check the concentrations in your ADS buffer, as they seem to give a 10x hypertonic solution.

Line 107-110: The M1 buffer has additives to a total of 102.5%. Perhaps adding glutamine, pen-strept and NCS increases the total volume? Please adjust the percentages accordingly.

Line 150: “Bay 60-7559” should read Bay 60-7550

Line 157+169+333: insert space between numerals and unit of measurement

Line 174: This reviewer assumes that the statistical analysis applies to the entire manuscript and not only the qPCR. Please insert a new paragraph termed “statistics data” or something similar.

Line 429+430: -1 in superscript

Supp page 5: should read “…by PDE8 is analogical.”

Author Response

Reviewer #2

The manuscript of Brescia et al. describes the development of three novel targeted sensors for the cyclic nucleotides cGMP and cAMP (PM-cGi500, OMM-cGi500 and PM-H187). These, in addition to three previously described sensors were expressed in neonatal left ventricular cardiomyocytes from C57BL mice. The sensors detected cGMP or cAMP in the bulk cytosol, at the plasma membrane and at the outer mitochondrial membrane in the basal and cell-stimulated state. In addition to this, the activity of several phosphodiesterases (PDEs) in the different subcellular compartments were analyzed in the basal and cell-stimulated state. The findings in normal mice were compared to a model of Duchenne Muscular Dystrophy termed mdx mice. Duchenne is often a result of functional loss of dystrophin that disrupts in sarcolemma integrity, and in cardiomyocytes to disruption of Ca2+ homeostasis, decreased NO production and mitochondrial signaling.

In the mdx model, the authors show that basal cGMP levels are decreased, specifically around the plasma membrane and that this might be a result of increased local PDE activity of certain isoforms. Beta-adrenergic receptor activation increases cAMP levels and reveals a different distribution of PDEs in the mdx model. These data are nicely modeled to a two-compartment model showing how the complex distribution of PDEs modulate cAMP levels at the outer mitochondrial membrane.

The manuscript is well written and contains detailed information on how soluble GC and beta-adrenergic receptor signaling is modified in the mdx model. The manuscript provides significant advances in tools for localized signaling of cyclic nucleotides and also significant advances in understanding the localization of both stimulators and PDEs in the plasma membrane and outer mitochondrial membrane of cardiomyocytes.

We thank this reviewer for the positive comments on our work.

Major concerns:

The authors need to better describe the rationale for choosing to construct plasmamembrane-localized sensors. There is a larger rationale to do this for the cAMP-sensor, since beta adrenergic signaling occurs at the plasma membrane, but it is less clear to this reviewer that soluble GC is restricted to the plasma membrane (about 2/3 is present in the cytosol DOI: 10.1161/CIRCRESAHA.111.259242). Please provide a rationale for a PM-localized cGMP sensor. Determining cGMP at the plasma membrane from stimulation with natriuretic peptides would provide a better localized response than from sGC. Please provide information or data showing localization of sGC, and also if this localization is changed in the MDX model. This is particularly important to determine if the reduced sGC seen in the PM sensor (and not in the cGi500 or OMM sensors) is due to altered localization of sGC.

We apologise with this reviewer for not providing sufficient justification for the choice of targeting the cGMP reporter to the plasmalemma. By choosing to target the cGMP sensor to the plasmalemma we did not mean to imply that sGC is restricted to this site. In fact, the activity of cytosolic sGC  is monitored in our study using the cytosolic cGi500. However, as remarked by this reviewer, a significant amount of sGC localizes to the plasmalemma and, given the role of dystrophin in anchoring nNOS at this location, it could be expected that in mdx mice, where the lack of dystrophin results in displacement of nNOS, the reduced amount of NO generated at the plasmalemma would result in reduced cGMP. To make this clearer, we have included a sentence on p. 10, second paragraph, including references reporting the localization of sGC to the plasmalemma.

We agree with this reviewer that it would be of interest to establish whether the reduced cGMP response detected at the PM is the consequence of altered localization of sGC. Unfortunately, we were not able to perform western blots on subcellular fractions given the limited amount of stored tissue and the inability to generate fresh lysates given current COVID-19 related restrictions in our animal facility (see also comments to reviewer #1). However, we were able to test the level of expression of sGC in whole cell lysates, which resulted to be elevated in mdx compared to control (new Supplementary Fig 3), indicating that, overall, there is not a decrease of sGC enzyme in the dystrophic myocytes. We also include in the revised manuscript a set of data that we believe helps explaining the reduced cGMP levels at the PM of mdx myocytes. As shown in the new Fig 2D, when we treated the myocytes with cinaciguat, a compound that, unlike NO-donor, can activate the oxidized form of sGC, we found that the cGMP response at the PM of mdx cells is significantly higher than in control cells. In contrast, when detected with the cytosolic sensor, the cGMP response to cinaciguat is not significantly different in mdx compared to control cells. These data indicate that in mdx myocytes the reduced cGMP signal at the PM is likely not due to relocalisation of sGC but to a higher proportion of oxidized enzyme, that appears to be selectively at this site.  

We did not perform experiments where cGMP is generated by activation of the particulate form of GC as pGC this is not influenced by NO levels and therefore is not directly relevant to our question.

Image quality: The fluorescent images in the manuscript version sent from the journal to this reviewer was of poor quality with a high amount of distortion. I suspect that this is due to technical difficulties in the manuscript sent to the reviewers and that the printed version is of higher quality.

We apologies for the poor quality of the images. We believe that, indeed, this is a consequence of the pdf conversion on submission and we are confident that the original, high resolution .tif images are of appropriate quality.

Sensor localization: As judged by the images provided, the authors need to determine if there is a nuclear localization of PM-cGi500, PM-H187 and OMM-H187? I noticed that the authors included DAPI in their methods section but not in their fluorescent images. Could DAPI be included to judge nuclear localization of these sensors? If there is a high degree of nuclear localization, was this included in the region of interest analyzed for FRET? In addition, possibly due to the reduced quality of the images received, is there significant overlap between OMM-cGi500 and the mitotracker used? A Pearson correlation coefficient (or similar) would be helpful to judge the co-localization.

Since the MDX model alters localization of signalosomes, is the localization of the localized sensors preserved in the MDX model? This is particularly important for the OMM sensors, since the authors refer to mitochondrial biogenesis being altered in DMD.

The DAPI staining was included in the methods section by mistake. We apologise for this and we have now removed the relevant text. In a small number of cells there is some mistargeting of the sensor to the perinuclear region. In those cells, we confirmed that, excluding the nucleus and perinuclear region form the ROI within which the FRET signal is measured does not significantly affect the ratio values. To more clearly illustrate appropriate targeting of the reporters, we have included in the revised manuscript Pearson’s correlation coefficient measurements for both PM- and OMM-targeted sensors. As shown in the new Supplementary Fig 1, coefficient values are between 0.65 and 0.83, indicating a good degree of co-localisation. In addition, our analysis shows that the degree of co-localisation of the targeted reporters, both at the PM and at the OMM, is not affected in mdx cells.

Traces: There is a lack of representative FRET traces to show the behavior of these newly developed sensors. Please provide such representative traces.

We now show representative traces for all sensors in the new Supplementary Fig 2

Saturation and dynamic range of localized sensors: In some of the sensors, such as the PM-cGi500, there appears to be a lower FRET change in the presence of stimulator than in the absence (e.g. Bay60 and PF044 for MDX). There could be several reasons for such a discrepancy, including different source of the basal cGMP compared to the sGC-stimulated or it could be due to saturation of the sensor and therefore not achieving higher FRET changes. Determining this is important, since the authors have previously shown that some sensors yield a decreased efficacy in the presence of a targeting domain (DOI:10.1038/ncomms15031). Could the authors provide data that the localized sensor are saturated after for example sGC-stimulated and PDE2-inhibited cells in the MDX model (PM-cGi500) by subsequent addition of either IBMX or natriuretic peptides? The H187 sensor seems to provide much lower FRET changes in the targeted sensors upon PDE-Inhibition despite similar increase in isoproterenol-stimulated FRET changes. Is this due to reduced efficacy of the localized sensors (could be determined with addition of forskolin) or reduced localization of PDEs in either the PM or OMM?

We thank this reviewer for raising this important point. In fact, after activation of sGC and application of any of the PDE family-selective inhibitors, the PM-cGi500 sensor is not saturated, as subsequent addition of IBMX further increases the signal. Although we don’t have data to support a mechanism responsible for this, one possibility is that the smaller response to PDE inhibitors after sGC activation may be the consequence of cGMP-dependent regulation of PDE activity or cGMP-dependent regulation of pGC activity. We completely agree with this reviewer that, for correct interpretation of results it is important to establish the dynamic range of the targeted sensors. To address this point we now include in the new Supplementary Fig 2, data showing the minimal (zero cyclic nucleotide) and maximal (1mM cyclic nucleotide) FRET change for cGi500, PM-cGi500, OMM-cGi500, H187 and PM-H187, as obtained by microinfusion via a glass pipette. Previously published data show that the dynamic range for OMM-H187 is similar to cytosolic H187 (doi: 10.1073/pnas.1806318115). As we previously reported (DOI:10.1038/ncomms15031), the data show that fusion of a targeting domain to a FRET reporter often affects its dynamic range. The data also confirm that the response obtained by sGC activation and PDE inhibition are not saturated. In the case of H187, the dynamic range of the sensors shows that the response to PDE inhibitors in the presence of agonist recorded at the PM and OMM is not saturated (as confirmed by application of forskolin and IBMX, as shown by representative curves in Supplementary Fig 2). Although the dynamic range of the PM-H187 is smaller than for the cytosolic sensor, the cAMP response relative to the maximal response at saturation is still smaller at the PM and OMM than in the cytosol. As mentioned above, the OMM-H187 is similar to H187. Therefore, the data indicate that the difference in cAMP response is to be attributed to different PDE activity at PM and OMM. To give an example, the response to PDE4 inhibition in activated mdx cells is 65% of max in the bulk cytosol but only 20% of max at the PM and 30% of max at the OMM; the response to PDE8 inhibition, on the other hand, is 10% of max in the bulk cytosol, 3% of max at the PM, but 15% of max at the OMM.

Selectivity of sildenafil: The IC50 of sildenafil for PDE5 was found to be ~7 nM, whereas the IC50 for PDE1 was ~200 nM (DOI: 10.1124/jpet.109.154468.). Since the authors uses ~1000 the IC50 of PDE5 (10 µM), which would also inhibit PDE1, does this change the interpretation of involvement of PDE5 vs. PDE1 in the PM-cGi500 sensor measurements? This would be particularly important if there is an increase in expression of PDE1 in the MDX model (which was not analyzed in Supplementary Figure 1).

We are aware that at the concentration used sildenafil also partially inhibits PD1 and PDE9. However, as the aim of our study was not to establish the contributions of individual PDEs to local cyclic nucleotide signalling and we used PDE inhibitors simply as a means to manipulate local cAMP and cGMP levels to allow a comparison between control and mdx, we considered this not to be an issue (see also response 1) and 2) to reviewer #1). In fact, selective PDE1 inhibitors that are not fluorescent, and therefore can be used in combination with FRET imaging, are not commercially available and the higher dose of sildenafil used in the study allows us to also target PDE1. We appreciate however the point raised concerning the interpretation of the role of PDE5 in the absence of any assessment of PDE1 expression. In the revised manuscript we now include mRNA data for PDE1A and PDE1C, the two PDE1 isoforms expressed in mouse heart. As shown in the revised Supplementary Fig 4, both PDE1 A and C, and PDE9, which is also partly inhibited at 10 μM sildenafil, are expressed at low and comparable levels in control an mdx myocytes.  We therefore conclude that it is unlikely that the differences we detected on application of sildenafil are the result of PDE1 overexpression in the dystrophic cells. We however agree that the reduced selectivity of sildenafil at the concentration used in this study should be acknowledged explicitly, and we do so in the revised manuscript on p 10, second paragraph

The authors have performed a large amount of work to include several variables (two models, PDE-inhibition in basal and stimulated cells) that creates several permutations. Are all the cells included in the measurements of basal FRET and stimulated FRET? If not, is there high enough statistical power to determine some of these differences? Depending on whether there are 4 cells, an average of 4 biological repeats or even 16 cells from 4 biological repeats, the statistical analysis of changes in Figure 4D should be performed again, as the error bars are greatly overlapping, while a p<0.01 is stated.

All the cells were included into the measurements of basal and stimulated FRET (multiple cells from the same biological replicate were averaged and the average of each biological replicate was used for statistical analysis). In revising the figures, we have however realized that the error bars in Figure 4D did not represent the correct SEM. The figure has been amended accordingly. 

In some of the experiments, a higher basal FRET was coinciding with a larger PDE activity of some PDEs (eg. PDE2 in cGi500, PM-cGi500 and OMM-H187, PDE9 in PM-cGi500 and PDE8 in OMM-H187). Did I miss a comment on this in the discussion section?

It is true that in two instances (i.e. PDE2 in cGi500 and OMM-H187) the basal cyclic nucleotide level is higher despite a higher PDE activity. We think this is not an inconsistency, as a higher level of PDE activity is still compatible with higher cyclic nucleotide level. In fact, this finding suggest that the amount of cyclic nucleotide generated is far more than detected, as it is partially degraded by a more pronounced PDE activity.  In the case of cytosolic cGMP, for example, were the activity of PDE2 is higher in control compared to mdx, the basal cGMP level would be even higher in control than detected. The same applies to the cAMP response at the OMM. In the latter case, the higher basal cAMP in controls despite higher PDE2 activity may be also compensated by lower PDE8 activity at this site. In all other cases listed by the reviewer, (PDE9 in PM-cGi500, PDE8 in OMM-H187 and PDE2 in PM-cGi500) a lower basal cyclic nucleotide levels corresponds to a higher PDE activity.

The abstract does not contain a summary of your findings. The authors must indicate their findings in the abstract.

This has been addressed

Illustration: It would be very helpful for the reader with an illustration of the localization of different PDEs and how these are altered in the MDX model.

 We have now included a schematic (Supplementary Fig 7) that illustrates the localization of the PDEs in different subcellular compartments and how their contribution to local CN levels is affected in basal conditions in mdx myocytes

Minor:

Figure legends: There is a large amount of information missing in figure legend 1E and 5B.

Amended

Figure caption: In PDE experiments, it might be difficult for the reader to understand the difference between inhibiting PDEs in the absence (basal) and presence of pre-stimulation without reading the figure legend. Please indicate on the panels if it is basal or pre-stimulation of a cAMP/cGMP elevating drug.

We thank the reviewer for this suggestion. The relevant figures have been amended accordingly.

Error bars: Please keep error bars uniform. In some panels (e.g. Figure 1C), there is a mix between upward and error bars in both directions.

This has been addressed

The authors did not see much cGMP using the sGC stimulator and performed all experiments in the presence of the NO donor SNAP in addition to sGC stimulator. Although the sGC stimulator enhances NO-stimulated sGC, I wondered whether the authors compared SNAP alone with SNAP + Bay41?

Unfortunately, the FRET change detected after application of SNAP only resulted to inconsistent. We did not pursue optimization of this treatment as the combination of Bay41 and NO donor showed synergistic response and reliable FRET change. Therefore, all experiments were performed with the combined treatment.

Supplementary Figure 3: To aid the reading and understanding of this figure, it might be useful with a heading for each panel. In line with this, the heading “Perturbed model” seems difficult to understand. Since this model assumes that there is no alteration of PDE8 activity in the MDX model, altering the heading to include this might be useful.

The figure has been amended and now incorporates this reviewer’s suggestions (new Supplementary Fig 8)

Reference list: please include all authors or insert “et al.” after the 10 authors you have listed according to the journal’s guidelines. Please update formatting of reference 41.

References have been amended

Line 52: 2+ in superscript

Amended

Line 90: insert a period after “isolation”.

Amended

Line 97-98: please check the concentrations in your ADS buffer, as they seem to give a 10x hypertonic solution.

Line 107-110: The M1 buffer has additives to a total of 102.5%. Perhaps adding glutamine, pen-strept and NCS increases the total volume? Please adjust the percentages accordingly.

We apologise for these mistakes. Correct values have now been introduced

Line 150: “Bay 60-7559” should read Bay 60-7550

Amended

Line 157+169+333: insert space between numerals and unit of measurement

Amended

Line 174: This reviewer assumes that the statistical analysis applies to the entire manuscript and not only the qPCR. Please insert a new paragraph termed “statistics data” or something similar.

Amended

Line 429+430: -1 in superscript

Amended

Supp page 5: should read “…by PDE8 is analogical.”

Amended

Round 2

Reviewer 1 Report

The authors have taken into accounts all comments made by the reviewer and precise their objectives, furthermore  have reported supplementary data to assess their data.  They have included some suggested  data expression by adding new experimental mRNA data. Although I am not fully convinced by their sol experimental approach, I recognize that this an interesting work opening new therapeutic approaches and I thank the authors for their new synthetic  supplemental figure 7, which merits to be included in the manuscript.

In fact, ref [44]  Tetsi et al. is include page 10 instead of page 13 according to author response.

Nevertheless, it remains- some problems in reference numbers that must be revised:

In the first version ref 21-24 were included in technics that were before results, however in the last version, since reference numbers are associated with their appearance in the text after the Conclusion their number must be consequently changed as well as all the reference numbers included from Results data till  the Methods

Author Response

In the first version ref 21-24 were included in technics that were before results, however in the last version, since reference numbers are associated with their appearance in the text after the Conclusion their number must be consequently changed as well as all the reference numbers included from Results data till  the Methods

We apologise for this error. All references have been corrected and they now follow a sequential number according to their first appearance in the text.

Reviewer 2 Report

The authors have taken all my comments into consideration and added data and illustration to support their data and conclusions. All major concerns that I raised are found in the revised manuscript.

I commend the authors for including a schematic illustration of their findings. From such a figure, it will be difficult to render whether basal or stimulated PDE activities are altered in the mdx model. That aside, please indicate in the legend that the black arrows indicate changes in the mdx mice compared to controls. I would hope that the authors would include the newly added illustration (Supplementary Figure 7) in the main file.

Thank you for clearing up sensor localization in mdx, western blots of sGC and representative traces.

Minor:

Line 98-9: Redundant wording.

Author Response

I commend the authors for including a schematic illustration of their findings. From such a figure, it will be difficult to render whether basal or stimulated PDE activities are altered in the mdx model. That aside, please indicate in the legend that the black arrows indicate changes in the mdx mice compared to controls. I would hope that the authors would include the newly added illustration (Supplementary Figure 7) in the main file.

We have now moved Supplementary Fig 7 to the main text, as Figure 7. We have also included in the legend specific reference to the arrows and what they represent (line 298-299)

Line 98-9: Redundant wording.

The repetition has now been deleted.